

# Beyond support: exploring the dynamic and static biomechanical changes induced by preventive ankle taping: a novel cross-sectional study

María Bravo-Aguilar , Vanesa Abuín-Porras , María Blanco-Morales , Carlos Romero Morales , Jaime Almazán-Polo , Jorge Hugo Villafañe and Lorena Canosa-Carro

Department of Physiotherapy, Faculty of Medicine, Health and Sports, European University of Madrid, Villaviciosa de Odón, Madrid, Spain

Corresponding authors
María Bravo-Aguilar,
maria.bravo@universidadeuropea.es
Carlos Romero Morales,
carlos.romero@universidadeuropea.es

## ABSTRACT

**Introduction**. In sports, 80% of all ankle injuries are sprains of the external compartment. Functional bandages are usually used preventively, specially in individuals with a history of lateral ankle injuries. To this day, the actual benefits of such taping remain unknown as important modifications are introduced in the ankle biomechanics.

**Objective**. The aim of the present study is to describe the biomechanical processes underlying these effects, such as modification during stance times, balance, contact surface and maximum and average pressures in the rearfoot, forefoot and midfoot, using a sprain preventive taping for the external ankle compartment.

**Methods**. An observational, analytic, cross-sectional study was designed. Data from static and dynamic plantar pressures with a pressure platform and balance data assessed with the Y Balance Test (YBT) were analysed in 50 participants (age = $21.00 \pm 2.34$ years, weight = $71.11 \pm 13.12$ kg, height = $1.75 \pm 00.9$ m, BMI = $22.94 \pm 2.50$ kg/m2, foot size = $41.60 \pm 3.00$) with and without preventive functional taping for lateral ankle sprain (LAS).

**Results**. A statistically significant decrease in YBT was observed in the taped participants toward anterior ($p = 0.001$) and posterolateral ($p = 0.005$) motion. On the static measures at the pressure platform, an increase in peak pressure at the midfoot ($p = 0.001$), a decrease in the maximum pressure in the forefoot ($p = 0.003$) and a decrease in the contact surface in the rearfoot ($p = 0.003$) were recorded. Dynamic measures at the pressure platform analysis showed a statistically significant decrease in contact surface at the rearfoot ($p = 0.001$), an increase in mean pressure in both the midfoot ($p = 0.044$) and forefoot ($p = 0.001$) and a significant decrease in velocity in the forefoot ($p = 0.003$).

**Conclusions**. In conclusion, we observed that ankle taping led to increased peak pressures in the midfoot and decreased maximum pressures in the forefoot, indicating a shift in load distribution within the plantar surface. Simultaneously, a significant reduction in the velocity at the forefoot during dynamic tasks suggests that taping may alter natural gait dynamics, potentially affecting movement efficiency and stride characteristics. In addition, the application of ankle taping significantly altered balance, as evidenced by a decrease of YBT scores anterior and posterolateral directions. Prophylactic taping in patients with no prior history of LAS is not recommended.

## INTRODUCTION

Preventive strategies for athletes with a history of ankle injuries often include multiple approaches to minimize the injury and re-injury risk. These strategies usually are based on the use of insoles or footwear designed to improve the lower limb and foot biomechanics, educational programs to raise injury awareness, proprioceptive training focused on enhancing motor control of the ankle and foot and the application of functional taping during sports activities to improve joint stability (*Kerkhoffs et al., 2001*; *Janssen et al., 2017*).

Functional ankle taping is frequently employed by athletes and medical staff as a preventive method against injuries. Several studies have assessed the biomechanical effects of ankle taping, particularly its capacity for limiting the joint movement, and have often compared it to orthopedic devices. While several authors report similar efficiency to the movement limitation between the two approaches, while other authors defend a greater limitation for the orthopedic devices, leading to improve injury prevention outcomes (*Kemler et al., 2011*). The widespread use of ankle taping is attributed to its comfort, quick application and perceived increase in joint stability, factors that contribute to the acceptance of this method in athletic environments (*Cooke et al., 2009*; *Kemler et al., 2011*).

Currently, it exists a generalized acceptance of the use of preventive ankle taping due to the the high prevalence of lateral ankle sprain (LAS), accounting for 80% of all ankle injuries (*Woods et al., 2003*). In this line, 70% of LAS patients have been reported to experiment recurrent injuries, leading to the development of chronic ankle instability (CAI) (*Anandacoomarasamy & Barnsley, 2005*). Residual symptoms are frequent for months or even years following LAS.

Several studies emphasized the importance of appropriate rehabilitation programs after a LAS episode to warrant proper healing context for the injured ligaments, allowing them to endure the tensile forces generated during ankle movements (*Denegar & Miller, 2002*). There is ongoing debate about the timeframe for achieving mechanical stability after an ankle injury, with some studies recommending a recovery period of at least 6 weeks to 3 months before returning to sports. However, instability may persist for up to a year following LAS (*Tohyama et al., 2003*; *Hubbard & Hicks-Little, 2008*).

The biomechanical assessment of ankle and foot pressures and load distributions using pressure platforms has emerged as a highly reliable and non-invasive method, employed in research and clinical practice settings. This technology is particularly useful due to their reproductible data which are essential for evaluating biomechanical interventions, such as preventive ankle taping and their effects on foot function and injury or re-injury risk. As a result, pressure platform analysis has become an indispensable tool for both diagnosis and managing conditions related with foot and ankle biomechanics, while also provides data for the development of targeted rehabilitation strategies and preventive programs (*Cuevas-Martínez et al., 2023*).

While some evidence suggests that the protective impact of ankle taping is particularly pronounced in individuals with a history of LAS, the actual benefits of such taping in preventing initial injuries or enhancing ankle dorsiflexion, stability, and balance remain unknown. Additionally, recent research indicates that ankle taping may inadvertently increase the risk of knee and ankle injuries due to altered lower limb biomechanics (*Romero-Morales et al., 2023*; *Romero-Morales et al., 2020*; *Romero-Morales et al., 2023*). These findings challenge the effectiveness of ankle taping for injury prevention in healthy populations and call for a critical reassessment of its utility, especially in healthy athletes without prior ankle injuries. Because the use of prophylactic taping is a fairly extended practice in sports even though its effectiveness has not been proven for healthy populations or athletes that had not have a previous LAS history (*Handoll et al., 2011*), nor has it the mechanisms of why it might be effective as a preventive measure for LAS the present study aims to explore the acute effects of preventive ankle taping on plantar pressure distribution and balance in healthy individuals without previous injuries. the Authors hypothesize that taping alters dynamic and static conditions, potentially shifting the load distribution and balance within the foot, which may influence the risk of injuryost of the reviewed literature focuses on specific populations, such as athletes or sports people recovering from injuries, who may already present biomechanical alterations. To the best of our knowledge, this is the first study to analyze the biomechanical impact of taping on both dynamic balance and plantar pressures under static and dynamic conditions in a healthy population.

## METHODS

### Study design

This cross-sectional, analytic, observational study was designed adhering to the Strengthening the Reporting of Observational Studies in Epidemiology (STROBE) guidelines, was performed between June 2023 and October 2023 at the Physiotherapy Research Laboratory of the European University of Madrid.

### Participants

Fifty healthy students from University Europea de Madrid volunteered for the study Table 1. The recruitment was carried out by a physiotherapist with more than 10 years of experience.

The exclusion criteria for participation in the current study included students with dermatological disorders or allergy to bandage, having undergone lower limb surgery, having had a lateral ankle sprain in the last three months.

All the participants ($n = 50$, age $= 21.00 \pm 2.34$ years, weight $= 71.11 \pm 13.12$ kg, height $= 1.75 \pm 00.9$ m, BMI $= 22.94 \pm 2.50$ kg/m2, foot size $= 41.60 \pm 3.00$) completed the study protocol (Tables 1 & 2).

### Ethics

The study was approved by the Ethics Committee of the European University of Madrid (CI Code: 2023-422) in accordance with the Declaration of Helsinki. The informed consent form was signed by all the participants before the beginning of the study.

**Table 1  Descriptive data.**

| Measurements | Male ($n = 29$) | Female ($n = 21$) | Total ($n = 50$) |
|---|---|---|---|
| Age, y | $21.51 \pm 2.54$ | $20.80 \pm 2.01$ | $21.00 \pm 2.34^{\dagger}$ |
| Weight, kg | $79.01 \pm 11.17$ | $59.95 \pm 5.07$ | $71.11 \pm 13.12^{*}$ |
| Height, m | $1.81 \pm 0.06$ | $1.66 \pm 0.05$ | $1.75 \pm 0.09^{*}$ |
| BMI, kg/m$^2$ | $23.86 \pm 2.58$ | $21.67 \pm 1.75$ | $22.94 \pm 2.50^{*}$ |
| Foot size | $43.82 \pm 1.73$ | $38.66 \pm 1.13$ | $41.60 \pm 3.00^{*}$ |
| Dominance side (right/left) | 26/3 | 16/5 | 42/8 |

**Notes.**
*Mean $\pm$ standard deviation (SD) was applied.
$^{\dagger}$Median $\pm$ interquartile range (IR) was used.

**Table 2  Plantar pressure platform variables with and without ankle taping.**

| Measurement | No taping | Taping | P-value |
|---|---|---|---|
| YEB A (cm) | $83.00 \pm 13.25^{\dagger}$ | $81.22 \pm 8.51^{*}$ | $0.001^{\ddagger}$ |
| YEB L (cm) | $78.50 \pm 17.25^{\dagger}$ | $78.50 \pm 15.25^{\dagger}$ | $0.626^{\ddagger}$ |
| YEB R (cm) | $78.00 \pm 14.75^{\dagger}$ | $76.44 \pm 11.92^{*}$ | $0.005^{\ddagger}$ |
| St_ MaxPress R (g/cm$^2$) | $745.72 \pm 128.55^{*}$ | $759.02 \pm 112.71^{*}$ | $0.051^{**}$ |
| St_MaxPressM (g/cm$^2$) | $533.50 \pm 125.96^{*}$ | $558.58 \pm 120.37^{*}$ | $0.001^{**}$ |
| St_ MaxPress F (g/cm$^2$) | $575.14 \pm 106.56^{*}$ | $530.80 \pm 106.83^{*}$ | $0.003^{**}$ |
| St_Cont_Sur_R (cm$^2$) | $32.56 \pm 5.78^{*}$ | $30.70 \pm 6.38^{*}$ | $0.003^{**}$ |
| St_Cont_Sur$^2$_M (cm$^2$) | $42.38 \pm 11.54^{*}$ | $41.64 \pm 10.36^{\dagger}$ | $0.354^{\ddagger}$ |
| St_Cont_Sur_F (cm$^2$) | $33.48 \pm 8.52^{*}$ | $31.80 \pm 6.57^{*}$ | $0.113^{**}$ |
| Dn_Cont_Sur_R (cm$^2$) | $33.04 \pm 5.08^{*}$ | $31.00 \pm 4.64^{*}$ | $0.001^{**}$ |
| Dn_Cont_Sur M (cm$^2$) | $54.37 \pm 12.70^{*}$ | $56.00 \pm 12.44^{*}$ | $0.287^{**}$ |
| Dn_Cont_Sur F (cm$^2$) | $43.13 \pm 9.08^{\dagger}$ | $41.82 \pm 8.99^{*}$ | $0.080^{\ddagger}$ |
| MPress _R (g/cm$^2$) | $841.00 \pm 248.00^{\dagger}$ | $884.00 \pm 329.50^{\dagger}$ | $0.580^{\ddagger}$ |
| MPress _M (g/cm$^2$) | $1113.77 \pm 331.98^{*}$ | $1229.46 \pm 326.96^{*}$ | $0.044^{**}$ |
| MPress _F (g/cm$^2$) | $1169.00 \pm 323.50^{\dagger}$ | $1174.00 \pm 408.50^{\dagger}$ | $0.001^{\ddagger}$ |
| MaxPress _R (g/cm$^2$) | $2153.60 \pm 504.08^{*}$ | $2251.60 \pm 519.64^{*}$ | $0.206^{**}$ |
| MaxPress _M (g/cm$^2$) | $2441.73 \pm 591.24^{*}$ | $2551.91 \pm 529.39^{*}$ | $0.193^{**}$ |
| MaxPress$^2$_F (g/cm$^2$) | $3036.22 \pm 671.97^{*}$ | $2962.37 \pm 645.83^{*}$ | $0.397^{**}$ |
| V_R (m/s) | $430.00 \pm 120.00^{\dagger}$ | $437.55 \pm 93.42^{*}$ | $0.753^{\ddagger}$ |
| V_M (m/s) | $622.88 \pm 75.27^{*}$ | $625.55 \pm 84.84^{*}$ | $0.785^{**}$ |
| V_F (m/s) | $644.00 \pm 77.64^{*}$ | $600.00 \pm 105.00^{\dagger}$ | $0.003^{\ddagger}$ |

**Notes.**
Abbreviations: A, Anterior; R, Right; L, Left; YBT, Y Balance Test; R, rearfoot; M, midfoot; F, forefoot. cm; St, Static plataform measures; Dn, Dynamic platform measures.; Con_Sur, Contact Surface; MPress, Mean pressure; MaxPress, Maximum pressure; V, Velocity; cm, centimetres; g/cm$^2$, grams per square centimeter; m/s, meters per second.
*Mean $\pm$ standard deviation (SD) was applied.
$^{**}$Student's $t$-test for paired samples was performed.
$^{\dagger}$Median $\pm$ interquartile range (IR) was used.
$^{\ddagger}$Wilcoxon signed-rank test was utilized.

## Outcome measures

Static and dynamic pressure analysis was carried out with the 1,600 sensors of the portable pressure platform Podoprint$^{\circledR}$ (Namrol Group, Barcelona, Spain) (*Izquierdo-Renau et al., 2017*).

The software allows you to assess parameters such as surface area measured un squares centimeters, the mean pressure measured in grams per square centimeter, the maximum pressure measured in grams per squared centimeters and the velocity measured in meters per second. For the measurement of velocity, the pressure platform records the exact moment when each part of the foot (forefoot, midfoot, and rearfoot) makes contact with the platform and when that contact ends. In this way, using the data on time and the distance covered, the platform calculates the displacement velocity. All the variables mentioned before were recorded for the forefoot, midfoot and rearfoot (*Cobos-Moreno et al., 2022*). For measuring these three different segments we took the rearfoot as the part of the foot comprised between the calcaneus and the transvers tarsal joint, the midfoot as the segment between the transverse tarsal joint and the tarsometatarsal joint and the forefoot as the segment from the tarsometatarsal joint to the distal phalange (*Matias et al., 2020*; *Takabayashi et al., 2017*).

For the YBT analysis the displacement was recorded in centimeters for the dominant foot. Three different measurements were assessed using separate analyses of the anterior, posteromedial, and posterolateral directions (*Plisky et al., 2009*; *Jagger et al., 2020*).

## Procedure

At the participant's arrival they were asked their age and their dominant foot; they were also weighted, and their height measured. After that, the participant was asked to stand still at the pressure platform to measure their static stance. Before recording their results for the static measurement, they were asked to take a few steps onsite, with their gaze straight ahead and their arms relaxed along the body. The participants were instructed to perform all tests barefoot to minimise the bias that could be derived from the use of footwear or insoles. This was performed by a physical therapist with more than 10 years of experience and extensive knowledge in gait biomechanics. For the dynamic pressure test, the platform was set up at 6 m in a 10 m long corridor at ground level. This distance minimizes the risk of gait adaptations and allows walking at a constant pace without the need to adapt the gait. All participants were given 5 min to practice before measurements were taken, to achieve as natural a gait as possible. The participant was asked to walk on the pressure platform until a clear picture of their gait pattern was obtained. For each participant several walks are necessary as the platform is 40 cm by 40 cm and therefore only one footprint can be recorded per walk; 4 complete plantar pressure images were obtained for each foot, with this number of images the results are considered reliable (*Cobos-Moreno et al., 2022*; *Hughes et al., 1991*; *Martínez-Nova et al., 2007*; *De Castro et al., 2014*). From the set of four recorded steps, the most accurate image of the dominant foot was evaluated to subsequently obtain the necessary data for the study.

Once their gait analysis was finished the participant was asked to take the YBT, which is a clinical adaptation of the Star Excursion Balance Test (SEBT) to assess their dynamic balance. The results from the test were recorded by a physical therapist with more than 5 years' experience who also explained how to perform the test to the participants and performed an instructional test. The participants were asked to perform three practice trials before taking the real test. The test consists in standing on the centre of the Y with
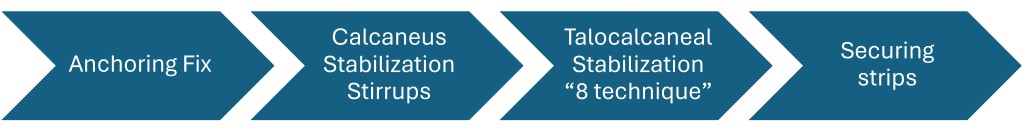

**Figure 1   Step-by-step taping Taping process.**

one leg while, with the other one, the participant tries to reach as far as possible into the three directions (anterior, posteromedial and posterolateral) while keeping their weight at the leg on the centre of the Y.

Afterwards the participants were asked to lay on a physiotherapy couch to get taped.

### Tape application

Ankle taping application was performed with a Strappal® 4-cm width non elastic tape from BSN medical (Essity, Stockholm, Sweden). The technique used was one of the most used ones by physical therapists, athletic trainers, coaches and athletes for ankle injury prevention or rehabilitation processes.

We ensured the accuracy of the bandaging and conditions, as they were performed by a therapist with over 15 years of experience in applying bandages (M.B.A). The ankle taping procedure was detailed in 4 steps (Fig. 1): (1) tape application technique anchoring strips: two Strappal® anchoring stripes, one at ankle height right above the medial and lateral malleolus and one at the forefoot. (2) Calcaneus stabilization: From those two anchoring stripes five tape stripes were applied to secure the calcaneus into a neutral position: three in an up position from the calcaneus to the ankle anchor and two of them from the calcaneus to the forefoot anchor. (3) Fixation of the talocalcaneal complex: with the ankle at 90-degree angle it was applied a figure-eight strip around the ankle joint with firm but comfortable tension. This strip should start at the forefoot, loop around the ankle, and cross back over to the forefoot, creating the figure called "eight pattern". (4) Close taping: finish the taping procedure applying additional strips without tension to ensure the entire bandage. These strips should cover and reinforce the previously applied tape (Fig. 2). It is considered mandatory to take into account the tension details: the tape should be applied with enough force to support the ankle joint without restricting blood flow or causing discomfort. This tape moderate tension is crucial for the anchoring strips and the figure-eight pattern. Thus, the ankle taping procedure should be firm but comfortable for the participants.

The order in which the procedure was carried out was to first have the participants measured without being taped at their arrival, then having the pressure and gait analysis recorded at the platform and finally, once they have performed the YBT, they were asked to get taped in order to retake every measurement again.

### Statistical analysis

Statistical Package for the Social Sciences (SPSS) version 26.0 (SPSS Inc., Armonk, NY, USA) was employed for the statistical analysis. Descriptive data were presented (mean and standard deviation for parametric data and median and interquartile range for non-parametric data) in Table 1. A normal distribution of quantitative data was assessed

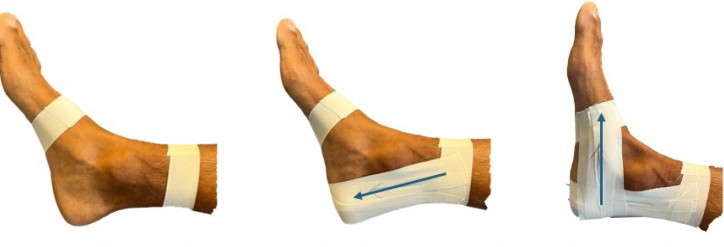

Two anchoring stripes, one at ankle height right above the medial and lateral malleolus and one at the forefoot.

From those two anchoring stripes five tape stripes were applied to secure the calcaneus into a neutral position: three in an up position from the calcaneus to the ankle anchor and two of them from the calcaneus to the forefoot anchor.

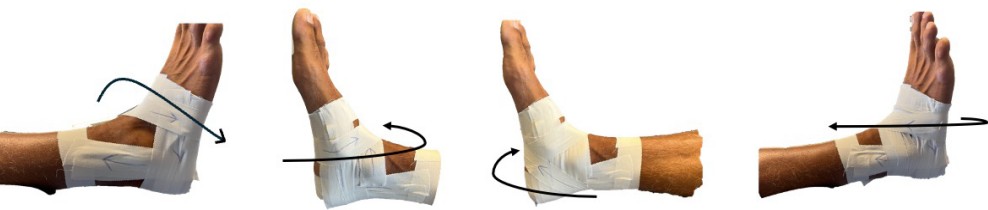

Fixation of the talocalcaneal complex: with the ankle at 90-degree angle. A figure-eight strip around the ankle joint with firm but comfortable tension. This strip should start at the forefoot, loop around the ankle, and cross back over to the forefoot, creating the figure called "eight pattern"

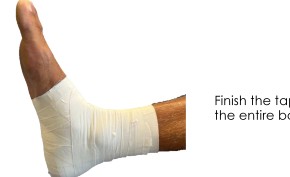

Finish the taping procedure applying additional strips without tension to secure the entire bandage

**Figure 2** **Visual taping application.**

by means of the Kolmogorov–Smirnov test. Differences between taping and no taping assessment were performed with Student-t test for related samples and Wilcoxon signed-rank, for parametric and non-parametric distributions.

## RESULTS

There were a total of 50 participants in the study, 29 males and 21 females (Table 1). A statistically significant decrease in distance was observed during the anterior YBT in participants while taped *versus* when not taped ($83.00 \pm 13.25$ and $81.22 \pm 8.51$ $p = 0.001$), as well as a decrease in distance during the posterior lateral YBT participants while taped *versus* when not taped ($78.00 \pm 14.75$ and $76.44 \pm 11.92$ $p = 0.005$). The static analysis at the pressure platform measurements, statistically significant changes were observed, an increase in the maximum pressure measured in grams per square centimeters in the midfoot ($533.50 \pm 125.96$ and $558.58 \pm 120.37$ $p = 0.001$), a decrease in peak pressure measured in grams per square centimeters in the forefoot ($575.14 \pm 106.56$ and $530.80 \pm 106.83$ $p = 0.003$) and a decrease in contact surface measured in square centimeters in the rearfoot ($32.56 \pm 5.78$ and $30.70 \pm 6.38$ $p = 0.003$).

The dynamic measurements made with the pressure platform showed statistically significant changes in the taped condition compared to the non-taped condition, a decrease in the contact surface measured in square centimeters in the rearfoot ($33.04 \pm 5.08$ and $31.00 \pm 4.64$ $p = 0.001$), an increase in mean pressure measured in grams per square centimeter in both the midfoot ($1,113.77 \pm 331.98$ and $1,229.46 \pm 326.96$ $p = 0.044$) and forefoot ($1,169.00 \pm 323.50$ and $1,174.00 \pm 408.50$ $p = 0.001$). A significant decrease in velocity measured in meters per second is also observed in the forefoot in the dynamic assessment ($644.00 \pm 77.64$ and $600.00 \pm 105.00$ $p = 0.003$). Table 2

## DISCUSSION

The present study evaluated the acute effects of preventive ankle taping on plantar pressure distribution and balance in healthy individuals. Our findings revealed significant changes in both static and dynamic conditions, highlighting the impact of taping on biomechanical parameters such as peak pressures, contact surface, and dynamic balance, as measured by the YBT. Those findings are aligned with previous studies, which reported similar results in pressure variables following the application of the taping (*Handoll et al., 2011*). The observed redistribution of plantar pressure could be explained by the compensatory mechanisms in response to restricted ankle mobility, particularly in the rearfoot, but this may increase the risk of overuse injuries in the midfoot and forefoot due to the increased concentration of localized stress.

*Migel & Wikstrom (2020)* conducted a review on different ankle stabilization methods in participants with LAS condition, highlighting the limited number of studies that evaluate the biomechanical effects of taping despite is widespread use in clinical practice. *Chinn et al. (2014)*, compared the effects of ankle taping with respect to no-taping on gait kinematics, demonstrating that taping enhances joint stability and for the limitation of the extreme range of motion movements. However, changes in plantar pressure, as observed in the present study, suggest that while ankle taping may stabilize the ankle joint, also lead to increase peak pressures in the midfoot and may cause uncontrolled load foot redistribution, which could increase injury risk.

In a pilot study developed by *Yen et al. (2018)*, the effects of the kinesiotape bandage on ankle kinematics using 3D motion analysis was assessed. Authors reported that the kinesiotape group exhibited a greater anti-inversion effect due to the properties of the materials. In the current study, similar effects were achieved, which may explain the comparable results. Our findings also showed a reduction in contact surface area in the rearfoot, which could indicate a more concentrated load potentially increasing the risk of overuse injuries or stress reactions. Moreover, the increased mean pressure in dynamic tasks observed in the midfoot and forefoot resulted in restricted rearfoot motion caused by the ankle taping. These findings were consistent with previous studies that suggest taping may alter natural load distribution patterns in the foot. In this line, the observed decreasing in forefoot velocity supports the idea that altered dynamic foot function was related with modifications in gait strategy and foot strike patterns, which could affect overall locomotor efficiency and potentially increase the risk of injury.

*Willems et al. (2005)* employed a pressure platform to compare non-injured individuals with those suffering from CAI, showing that CAI group exhibited a lateral shift in the center of pressure. In the present study, the taping appeared to promote a neutral position of the subtalar and talonavicular joint, which could partially explain the prevention effect of externally stabilizing the ankle that had been already described in scientific literature (*Quinn et al., 2000*; *Emery & Pasanen, 2019*; *Kaminski, Needle & Delahunt, 2019*).

Balance deficits are commonly observed in individuals with CAI (*Jaffri et al., 2020*), and preventive ankle taping plays and important role in addressing these deficits. *Trojian & McKeag (2006)* found the relationship between poor balance scores and higher incidence of ankle injuries. In the present study, a decrease in YBT performance in the anterior and posterlotaleral directions is consistent with previous findings, where lower YBT scores are strongly linked to a higher risk of lower limb injuries and impaired dynamic balance (*Trojian & McKeag, 2006*; *Mohammadi et al., 2024*; *Alkhathami, 2023*). Individuals with asymmetries in YBT are particulary vulnerable to injury, emphasizing the need for fitted interventions to correct these imbalances (*Smith, Chimera & Warren, 2015*). Consequently, a reduced ROM dorsiflexion observed in our study correlates with the stabilizing effect of taping on the talonavicular and subtalar joints, which helps to mitigate excessive movements of the talus during plantar flexion (*Bernier, Perrin & Rijke, 1997*; *Hertel et al., 1999*; *Meyer et al., 1988*). *Kerkhoffs et al. (2002)* remarked in a systematic review the effectiveness of the functional tape stabilization in extreme ankle joint movements. *Romero-Morales et al. (2023)* also reported a decrease in dorsiflexion following the preventive application of taping for LAS, which is consistent with our findings. The position of the tape stirrups plays an important role controlling the calcaneal adduction and supination (*Kaminski, Needle & Delahunt, 2019*). Additionally, these results support the use of preventive ankle taping to improve balance and joint stability (*Pollock et al., 2000*; *Zwiers et al., 2016*; *Chinn et al., 2014*).

The significant decrease in forefoot velocity observed during dynamic assessment suggest an altered foot function which is related with altered gait strategies or an increase in injury risk. Static platform analysis further revealed increased midfoot pressure, decreased forefoot pressure and reduced contact surface, consistent with previous findings about pressure redistribution due to ankle taping (*O'Sullivan et al., 2008*). The changes on the surface distribution may cause compensatory movements or altered load-bearing features associated with a decrease of the movement efficiency. Therefore, strategies focused on minimal biomechanical disruption should be considered (*Hopper et al., 2009*; *Sawkins et al., 2007*).

## Limitations and future lines

The present study has several limitations that need to be acknowledged. First, there is a possibility of gait adaptation by the participants to the pressure platform, which could potentially affect the naturalness of the gait patterns observed. Although measures were taken to minimize this by allowing practice time, the influence cannot be entirely ruled out. Second, the measurements were only taken immediately after the application of the bandage, which does not provide information on the longevity of the effects observed. It

would be beneficial for future research to examine the durability of these biomechanical changes after a period of athletic activity. Third, the study did not include functional tests for jumping movements, which are crucial activities in sports where lateral ankle sprains are common. This limitation could affect the applicability of the findings to real-world sports settings where dynamic and high-impact movements are frequent, so this study has to be considered only as a preliminary exploration of the ankle biodynamics behavior with the application of a functional bandage. Finally, the inclusion of only healthy participants can introduce a potential bias, as the results may not be applicable to situations where the biomechanic of the foot is already altered.

Future studies should aim to address the limitations noted by incorporating longer follow-up periods to assess the persistence of the taping effects through various phases of athletic activities. Additionally, including functional jumping tests could provide a more comprehensive understanding of the tape's effectiveness in dynamic sports scenarios. To obtain a fuller picture of the ankle and foot joint complex's behavior under taped conditions, it would also be beneficial to measure muscle activation patterns, particularly of the peroneal muscles and intrinsic stabilizing muscles of the foot, which play significant roles in ankle stability. Expanding the research to specific high-risk sports and tracking outcomes across different seasons could help in determining the role of taping in preventing chronic ankle instability and its efficacy in real-world sports applications.

### Clinical implications

The findings of this study suggest significant biomechanical alterations due to ankle taping that could influence clinical practices in sports medicine. While taping is shown to modify plantar pressures and gait dynamics, the implications for injury prevention and performance need careful consideration. Sports medicine professionals should weigh the benefits of ankle taping against potential alterations in natural movement patterns and the risk of compensatory injuries. Tailoring taping techniques to individual athletes' needs and monitoring their impact over time could optimize its protective effects while minimizing adverse outcomes. Furthermore, the impact of other rehabilitative techniques in ankle stability, such as dry needling (*Martínez-Jiménez et al., 2023*) should be explored both in clinical and research environments.

## CONCLUSIONS

The results of the present study showed that ankle taping alters biomechanical variables, including increased midfoot pressures and decreased forefoot pressures, suggesting a shift in load distribution. The reduction in forefoot velocity and YBT scores indicates changes in gait dynamics and balance. While taping enhances joint stability and may aid in preventing ankle injuries, these biomechanical alterations highlight the need for tailored taping strategies to ensure effective injury prevention without compromising movement efficiency and therefore the authors cannot recommend prophylactic taping for healthy subjects with no prior history of LAS.

### Funding
The authors received no funding for this work.

### Competing Interests
Carlos Romero Morales is an Academic Editor for PeerJ.

### Author Contributions
- María Bravo-Aguilar conceived and designed the experiments, performed the experiments, analyzed the data, authored or reviewed drafts of the article, and approved the final draft.
- Vanesa Abuín-Porras analyzed the data, prepared figures and/or tables, authored or reviewed drafts of the article, and approved the final draft.
- María Blanco-Morales conceived and designed the experiments, performed the experiments, authored or reviewed drafts of the article, and approved the final draft.
- Carlos Romero Morales conceived and designed the experiments, performed the experiments, analyzed the data, prepared figures and/or tables, authored or reviewed drafts of the article, revised the written manuscript, and approved the final draft.
- Jaime Almazán-Polo performed the experiments, analyzed the data, prepared figures and/or tables, authored or reviewed drafts of the article, and approved the final draft.
- Jorge Hugo Villafañe conceived and designed the experiments, analyzed the data, prepared figures and/or tables, authored or reviewed drafts of the article, and approved the final draft.
- Lorena Canosa-Carro conceived and designed the experiments, performed the experiments, analyzed the data, authored or reviewed drafts of the article, and approved the final draft.

### Ethics
The following information was supplied relating to ethical approvals (i.e., approving body and any reference numbers):

The study was approved by the Ethics Committee of the European University of Madrid (CI Code: 2023-422)

### Data Availability
The raw data is available in the Supplemental File.

### Supplemental Information
Supplemental information for this article can be found online at http://dx.doi.org/10.7717/peerj.18472#supplemental-information.

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
