# Peer review of "Beyond support: exploring the dynamic and static biomechanical changes induced by preventive ankle taping: a novel cross-sectional study"

_PeerJ, doi:10.7717/peerj.18472_

## Round 0.1 · original submission · Major Revisions

· Academic Editor

Major Revisions

Your paper presents information on the effect of ankle taping, which would be valuable for researchers and practitioners. We would like to see a major revision, with a point by point response to the comments of all three reviewers. Your revisions and responses to the comments of reviewer 1, who recommended rejection, will be especially important.

Reviewer 1 ·

Basic reporting

1. This study compared the plantar pressure data and functional test outcomes between the taping and non-taping conditions in various tasks. The results showed that ankle taping was effective in altering the functional test results as well as plantar pressure in different foot segments. However, the writing should be improved in both linguistic quality and the way the authors chose to present their research questions and results. I do not consider the current format can be accepted by the Journal. The specific comments are shown as follows.
2. One of the major issues of this manuscript is the research topic/purpose being unclear. Many factors, such as LAS, CAI, ankle instability, athletes, non-athletes, taping, and bandages, have been mentioned in the Introduction. However, it is not clear whether these are related to the topic of this study.
3. The Introduction section should be improved to present a clear description of the background information surrounding the research topic of this study. In addition, the research gaps are not clear, either. The last paragraph should also be enhanced to show the specific research topic (e.g., which population for analysis, which ankle taping, which kind of comparison conducted).
4. The Methods section should be improved to provide a clear flow of the protocols and steps conducted in this study.
5. According to the current study design, I find it difficult to relate the research results to any type of ankle injury, as only “healthy students” were included for analysis and the relationship between the measured variables and ankle injuries is not clear. In addition, this study primarily measured the plantar pressure under both standing and walking conditions. These are not closely relevant to the fierce movements in sports, wherein ankle injuries may be caused. The authors may reconsider the study design and research purpose.

Experimental design

This study conducted a series of tests to examine plantar pressure data and functional test scores. However, it is not clear whether these measured variables are related to ankle sprain, LAS, or CAI. Additionally, as mentioned by the authors that ankle taping has been widely used among athletes, I do not understand why “healthy students” were recruited for this study, instead of athletes at high risk. Lastly, it is not clear whether all participants went through both taping and non-taping conditions, or two groups each experienced one of the two conditions.

Validity of the findings

Table 1 and Table 2 can be combined.

Table 3. The abbreviations should be defined. Units should be provided if necessary. Yello highlights are not necessary. Was an independent or paired t-test performed in this study? I understand that the data may not be normally distributed or other assumptions were violated for t-test. The authors may consider transferring the data to meet the requirements for t-test. Alternatively, the authors may use non-parametric analysis (or other methods) to analyse the data.

Additional comments

1. Contemporarily, some argue against the use of the term ‘subject’ for human participants. Many researchers may prefer the term ‘participant’. The authors may consider which term would be more appropriate to use in a research article.
2. In both Abstract and main text, all abbreviations should be defined (and spelled out) before being used.
3. In the Abstract, more details should be included to provide a clearer description. For example, a clear study design should be presented (the characteristics of the participants, the taping conditions, the tasks/tests conducted, the measured variables, …).

Reviewer 2 ·

Basic reporting

This is a well-written manuscript with an important clinical message, and should be of great interest to the readers of Peer J. the dynamic and static biomechanical changes induced by preventive ankle taping is a prevalent condition so is very important in order to help readers about a better knowledge of this disability.


The topic of this study is trend, so it may be helpful to the readers.


On other hand plantar pressure and taping procedure so a right management of this technique could reduce the complication rates.



Introduction section is deep enough with and adequate focus that may help readers to improve knowledge about the topic. However authors should improve the stay of art, for example including references plantar pressure in sport I suggest to include this references include in the attached to complete this requirement
DOI:10.3390/bioengineering10070772




Methods section is right. Is clearly written the research methodology with a strong statistical .


Results section is clearly showed with an enough number of figures and tables that help to achieve a better understanding of this analysis.

Discussion section is well structured with different sections. Authors manage well the discussion leading a good comparison with the showed references.



However, author should discuss the possible influence of dry needle on plantar pressure as a rehabilitve treatment to include this references include in the attached to complete this requirement


DOI :10.1111/joa.13862

Experimental design

Methods section is right. Is clearly written the research methodology with a strong statistical .

Validity of the findings

Results section is clearly showed with an enough number of figures and tables that help to achieve a better understanding of this analysis.

Additional comments

This is a well-written manuscript with an important clinical message, and should be of great interest to the readers of Peer J. the dynamic and static biomechanical changes induced by preventive ankle taping is a prevalent condition so is very important in order to help readers about a better knowledge of this disability.


The topic of this study is trend, so it may be helpful to the readers.

Reviewer 3 ·

Basic reporting

Please refer 4. additional comments

Experimental design

Please refer 4. additional comments

Validity of the findings

Please refer 4. additional comments

Additional comments

1. There are instances of awkward phrasing and minor grammatical errors. For example, the phrase "the actual benefits of such taping in preventing initial injuries or enhancing ankle dorsiflexion, stability, and balance remain questionable" (Abstract) could be refined for clarity.
2. Same applies to the introduction as well. The transition between general background information and the study's specific aims could be smoother.
3. The introduction mentions "controversy regarding when mechanical stability after an injury is achieved," but does not elaborate on the nature of this controversy or cite specific studies. Providing more detailed examples and references would strengthen this argument.
4. Figure 1’s caption needs to be more descriptive.
5. The description of the taping technique could benefit from a diagram or more precise instructions, particularly regarding the tension applied to the tape.
6. The discussion lacks depth. For example, the practical significance of statistically significant differences, such as those observed in the YBT scores (please mention YBT is Y Balance Test first), should be discussed in terms of their impact on athletic performance and injury prevention.
7. Overall, the discussion needs additional content explaining the pre-post taping and its resulting change in metrics (for example, how it compares against literature, its implications, etc.).
8. Please consider talking about the bias that might be introduced by only university students.

---

## Round 0.2 · Minor Revisions

· Academic Editor

Minor Revisions

While the work is important, and executed well, the presentation still needs some improvements and clarifications. In your final revision, please address the remaining comments of Reviewer 1, and my Editor comments which are listed below.

1. Line 34, "tatic" should be "static".
2. Line 76: "pressures platform" should be "pressure platform".
3. Lines 76-77: This paragraph is very short and seems out of place, in the middle of the development of the clinical relevance of the work. Consider removing it, or placing this later in the Introduction, when presenting previous research that is relevant. Reviewer 1 also had important comments about the Introduction.
4. Line 80. The word "questionable" has a negative connotation, and should probably be replaced by "unknown".
5. Line 88: "Many" is grammatically incorrect. You can use "Much" or "Most".
6. Line 161. "participant's" should be "participants" (without apostrophe)
7. It is not my intention to proofread the whole manuscript for spelling and grammar, and I am sure there are more issues like these. Please make sure to have spelling and grammar corrected (MS Word can do much of that work) before resubmitting. Such errors make a bad impression on reviewers and editors.

Reviewer 1 ·

Basic reporting

I would like to thank the authors for revising the manuscript. However, the writing should be improved in both linguistic quality and the way the authors chose to present their research questions and results. I do not consider the revised manuscript to be accepted by the Journal.

While the revised Introduction section is lengthy and informative, it still makes me feel difficult to find the relevant research background or research gaps. For example, the comparison between taping and orthopaedic devices, the mechanism of LAS and CAI, and the LAS rehabilitation have been mentioned. However, their relationship with the main topic of this study is unclear. Furthermore, the Introduction usually covers the main topic of the study, which is acute taping effects on injury prevention. However, very limited lines have mentioned it. The Introduction section sometimes also mentions tested variables related to the main topic. However, previous studies have been mentioned regarding the effects on ankle injuries, biomechanics modifications, static and dynamic conditions, or balance test. Thus, the Introduction still needs some improvement.

The Discussion section, same as the Introduction, lists a lot of papers that may be relevant, but fails to provide a good discussion surrounding the topic and findings of this study.

The Conclusions section while showing the altered biomechanics variables caused by taping, seems not to mention the main topic of this study, ankle injury prevention.

Experimental design

The Methods section needs improvement. Experiment protocols and measurements are not clear and difficult to follow for readers. Detailed comments can be found under the “Additional comments”.

Again, as mentioned in the last round of revision, only “healthy” participants were included in this study. However, the Introduction talks a lot about ankle injuries and injury recurrence. This is not in line with the topic of this study. The authors are recommended to rethink the study design.

Validity of the findings

I still have some questions about the result presentation of this study. For example, for the variable “YEB A” in Table 2, the non-parametric analysis, Wilcoxon signed-rank test was used as indicated by its p value. In this case, median (IR) should be provided. However, the authors chose to present mean (SD) and median (IR) for both groups. This may not be common to see. The authors are recommended to refer to other papers using similar statistical tests to provide a better presentation of the results.

Additional comments

1. Line 66. There is an extra “participant”.
2. Line 115. What kind of velocity was measured?
3. Line 116. It is unclear how the three foot segments (e.g., forefoot, midfoot, rearfoot) were defined.
4. Line 122. Is the “force platform” referring to the same portable pressure platform or something else?
5. Line 124. Which “spot”? On the force/pressure platform?
6. Line 127. What is the demission of the platform? What is the position of the platform in the 10-m walk path? In each trial, how many steps can be captured using the platform?
7. Line 144. The 50 participants underwent both taping and non-taping treatments. It is unknown whether the two treatments were conducted a randomised order for each participant.
8. Line 147. A figure/photo for the bandaging would be much easier for readers to follow.
9. Line 171. This is not a complete sentence. Similar issues are also seen in many other parts of this manuscript.
10. Line 172. “the group that underwent preventive ankle taping” This phrasing could be misleading and readers may consider there were two groups. The authors may use “taping condition” and “non-taping condition” instead.
11. Line 181. Is the “dynamic platform” referring to the same pressure platform used for other tests, or another single platform used for this walking test?
12. Line 205. It is not clear which condition/group had a decreased contact surface. Similar issues for the following lines.

Reviewer 2 ·

Basic reporting

After to read carefully the R1 I consinder that authors have increased the manuscript in order to fit on the journal scope. Moreover, they attend the reviewer comments on the propert way

Experimental design

methods are well designed, futhermore, statistics are appropiate

Validity of the findings

Under my point of view achievement are very postive due to the fact it contribute to increase the knoweledge on foot diseases and sport

Additional comments

Author have adressed all my requeriments on the correct way

Reviewer 3 ·

Basic reporting

no comment

Experimental design

no comment

Validity of the findings

no comment

Additional comments

no comment

---

## Round 0.3 · accepted · Accept

· Academic Editor

Accept

Thank you, all reviewer comments have been addressed. I have attached a PDF file with my final suggestions for revision. I have removed some detail from the abstract, and reduced the significant digits in the reported results. This will help readers see the important information more easily. I used this as guidance: https://adc.bmj.com/content/100/7/608. Also I have corrected some grammatical and punctuation errors. If you agree with my changes, you don't need to do anything, I will make sure that the PeerJ staff receives that version. If you do not agree with my changes, please let us know.